# An Examination of the Complex Pharmacological Properties of the Non-Selective Opioid Modulator Buprenorphine

**DOI:** 10.3390/ph16101397

**Published:** 2023-10-02

**Authors:** Leana J. Pande, Rhudjerry E. Arnet, Brian J. Piper

**Affiliations:** 1Department of Medical Education, Geisinger Commonwealth School of Medicine, Scranton, PA 18509, USA; lpande@student.touro.edu (L.J.P.); rarnet03@som.geisinger.edu (R.E.A.); 2Touro College of Osteopathic Medicine, Middletown, NY 10027, USA; 3Center for Pharmacy Innovation and Outcomes, Danville, PA 17821, USA

**Keywords:** opioid use disorder, pain, opiate

## Abstract

The goal of this review is to provide a recent examination of the pharmacodynamics as well as pharmacokinetics, misuse potential, toxicology, and prenatal consequences of buprenorphine. Buprenorphine is currently a Schedule III opioid in the US used for opioid-use disorder (OUD) and as an analgesic. Buprenorphine has high affinity for the mu-opioid receptor (MOR), delta (DOR), and kappa (KOR) and intermediate affinity for the nociceptin (NOR). Buprenorphine’s active metabolite, norbuprenorphine, crosses the blood–brain barrier, is a potent metabolite that attenuates the analgesic effects of buprenorphine due to binding to NOR, and is responsible for the respiratory depressant effects. The area under the concentration curves are very similar for buprenorphine and norbuprenorphine, which indicates that it is important to consider this metabolite. Crowding sourcing has identified a buprenorphine street value (USD 3.95/mg), indicating some non-medical use. There have also been eleven-thousand reports involving buprenorphine and minors (age < 19) at US poison control centers. Prenatal exposure to clinically relevant dosages in rats produces reductions in myelin and increases in depression-like behavior. In conclusion, the pharmacology of this OUD pharmacotherapy including the consequences of prenatal buprenorphine exposure in humans and experimental animals should continue to be carefully evaluated.

## 1. Introduction and History

Buprenorphine was first derived from thebaine in 1966 and was subsequently characterized as a partial agonist at the mu-opioid receptor (MOR) [1]. The Committee on Drug Addiction primarily focused on morphine and looked for a way to ensure its multitude of uses without its addictive side effects in the 1920s. Buprenorphine was considered a part of the solution to the 20th century opium problem. Its agonist–antagonist pharmacological character was more fully characterized in 1972 and its potential as an addiction treatment recognized in 1979 [2,3]. Buprenorphine is a semi-synthetic and lipophilic drug. It has activity at all four major opioid receptors: MOR, kappa (KOR), delta (DOR), and the nociceptin receptor (NOR). Of the four main opioid receptors, three (MOR, DOR, and KOR) were identified in the 1960s and the opioid receptor like (ORL), currently and henceforth designated as NOP, was discovered in the 1990s [4]. In addition to its involvement in nociception, the KOR is widely expressed during prenatal and early postnatal periods including on progenitor, ependymal, and neuronal cells [5], which raises the possibility that a KOR antagonist such as buprenorphine could have an adverse impact on brain development. This may also apply to other (MOR/NOR) opioid receptors that are important for myelination [6]. The NOP is considered an atypical, low affinity receptor for opioid peptides [4]. Although marketed for analgesia and addiction treatment, early research subjects reported that buprenorphine was the “most reinforcing drug they had ever used” [2]. Injectable buprenorphine became commercially available in the US in 1981 [1]. By 1985, it was available in 29 countries [2]. Buprenorphine was originally considered a Schedule V narcotic in the US until 2002 when, after three attempts by the Drug Enforcement Agency, it was rescheduled as Schedule III [1,2]. US sales of buprenorphine have increased substantially. Buprenorphine was the most commonly used opioid by US veterinarians [7]. This may change as, by morphine mg equivalent, buprenorphine was only the seventh most common opioid in US veterinary teaching institutions [8]. From 2008 to 2019, buprenorphine distribution increased seven-fold to US pharmacies and five-fold to hospitals. The US Medicaid program spent 1.1 billion on buprenorphine in 2017 alone [8].

Although there have been many reviews about the pharmacological effects of buprenorphine [2,9,10,11], an examination of pharmacology educational materials [12,13,14,15] revealed a very simplified MOR-centric presentation of this drug. Therefore, the goal of this review was to examine the pharmacodynamics, as opioid neuropharmacology and therapeutics is an expanding field [16]. Other topical and related areas including pharmacokinetics, misuse potential, toxicology, and possible prenatal sequelae were also reviewed.

## 2. Pharmacokinetics and Pharmacodynamics

We now turn our attention to the pharmacokinetics (Table 1), pharmacodynamics, and their interaction for this semi-synthetic opioid. Many of buprenorphine’s pharmacokinetic properties explain its unique effects [17]. Buprenorphine’s metabolism follows non-saturable Michaelis–Menten kinetics [18] that furthers its analgesic effects [9]. There are two major metabolic pathways in buprenorphine’s metabolism. Buprenorphine undergoes *N*-dealkylation catalyzed by the hepatic cytochrome P (CYP) 450 (CYP P450-3A4) and glucuronidation, resulting in three major metabolites: buprenorphine-3-glucuronide (B3G), N-dealkylbuprenorphine, and norbuprenorphine-3-glucuronide (N3G). The CYP3A4 system metabolizes buprenorphine to norbuprenorphine through N-dealkylation of the cyclopropylmethyl group [9,19,20]. CYP3A4 is predominantly responsible for this, although CYP2C8 also contributes. The ratio of norbuprenorphine to buprenorphine in urine can provide an index of the recency of buprenorphine administration, the potential use of a CYP3A4 inducer, or the probability of buprenorphine “spiking”, involving submerging the film or tablet in urine in an effort to have a positive immunoassay result [21]. Some conclude that norbuprenorphine does not readily cross the blood–brain barrier ([9], although see [22]). Sheep are used for the similarity of their blood–brain barrier to humans. The large size of sheep contributes to their use in pharmacokinetics investigations. The peak concentration of buprenorphine was half that in the sagittal sinus relative to the arterial quantities, which indicated only intermediate permeability across the blood–brain barrier. In contrast, the peak concentration and time to peak concentration were very similar for samples from the sagittal sinus and arterial blood for norbuprenorphine. Also noteworthy was that the peak sagittal concentration of norbuprenorphine was over twenty-fold higher than that of buprenorphine in this species [23]. Among patients receiving buprenorphine/naloxone for two weeks, the twenty-four hour area under the concentration curve was equivalent for buprenorphine and norbuprenorphine [24]. Norbuprenorphine is commonly measured in urine analyses because of these high concentrations [25]. Norbuprenorphine and buprenorphine are both detectable in meconium, although norburprenorphine’s quantities were six-fold higher [26]. Another biological matrix that can provide an index of buprenorphine use is hair. The hair of pregnant women, and a small sample of their offspring, had measurable norbuprenorphine and buprenorphine [27]. Norbuprenorphine is then metabolized to N3G [17,28]. Other metabolites, including hydroxy buprenorphine and hydroxynorbuprenorphine, have been identified. CYP3A4 produces hydroxybuprenorphine [18]. The CYP3A4 activity varies between individuals and can be induced, resulting in wide differences in pharmacokinetics [18]. Buprenorphine is eliminated in the urine and in feces, accounting for one-third and two-thirds of the eliminated buprenorphine, respectively [18]. It is important to note that interactions between different drugs can occur due to the inhibitory effects on CYP3A4, which is often blocked by various medications For instance, when ritonavir, a potent CYP3A4 inhibitor, is administered alone, it can elevate the levels of both buprenorphine and norbuprenorphine without intensifying the adverse effects associated with buprenorphine [15]. For instance, when ritonavir, a potent CYP3A4 inhibitor, is administered alone, it can elevate the levels of both buprenorphine and norbuprenorphine without intensifying the adverse effects associated with buprenorphine [24]. As cannabis is a CYP3A4 inhibitor, a doubling of buprenorphine concentrations in regular cannabis users [29] may impact many patients that use cannabis for chronic pain [30]. Buprenorphine has an absolute contraindication in humans to not be combined with the antiretroviral atazanavir, the histamine (H_1_) blocker azelastine, the typical antipsychotic bromperidol, the irritable bowel agent eluxadoline, the sleeping sickness agent fexinidazole, the calcium antagonist flunarizine, the antibiotic fusidic acid, kratom, monoamine oxidase inhibitors, naltrexone, the H_1_ antagonist olopatadine, the anticholinergic orphenadrine, the antihistamine oxomemazine, the depressant paraldehyde, the opioid antagonist samidorphan, or the oncology agent thalidomide [31].

Buprenorphine has fewer drug–drug interactions than other opioids that are metabolized through CYP3A4 [1]. Further research needs to be conducted on buprenorphine’s drug interactions, as there is more information for methadone [36]. Buprenorphine alone may have a higher ceiling effect than typical MOR agonists, but in combination with benzodiazepines, it could result in a potentially life-threatening drug interaction due to sedation and respiratory depression properties. The mechanism of the respiratory depression is unclear [37,38]. There is some noted benefit to combining opioids with buprenorphine to produce sub-addictive analgesia [1]. In the postoperative setting, buprenorphine did not impair morphine analgesia (buprenorphine 0.4 μg/kg as an infusion and 0.15 μg/kg as the demand dose) [39]. Cancer patients with breakthrough pain receiving transdermal buprenorphine from 35–70 μg/h responded well to an oral morphine to transdermal buprenorphine ratio of 75:1 [40]. Additionally, those using high-dose buprenorphine for maintenance therapy did not need to be switched off this opioid for methadone, as the patients morphine responses were not different between the two groups [41].

Glucuronide metabolites of buprenorphine are biologically active, contributing to the pharmacology of the drug [9]. The glucuronidation rate is roughly the same for buprenorphine and norbuprenorphine in the liver and small intestine. *N*-dealkylation is one-hundred fold greater in the liver than in the small intestine [42]. Conjugated metabolites are excreted in bile and half the buprenorphine administered is eliminated in the feces [28]. In bile fistula rats, where the bile flows into a hollow structure when 0.6 mg/kg buprenorphine was administered intravenously, 75% of B3G and 19% of N3G were excreted in bile. In “linked rat models” or intact rats, approximately twice the amount of N3G was found to be excreted compared with B3G. There are differences in excretion due to first-pass effects in enterohepatic circulation [42]. It is deconjugated by the colon by bacteria, then reabsorbed [18].

Buprenorphine is an atypical opioid as a result of its receptor activity at the MOR [9]. Buprenorphine has shown activity at all four opioid receptors [3]. Buprenorphine dissociates from the MOR slowly, resulting in a slow onset and a long duration for the analgesic effects [3]. A 2002 review describes how the MOR partial agonist and KOR antagonist properties of buprenorphine have been well established but that there had been comparatively less research on DOR and NOR [11]. Although most opioids show activity at the MOR, DOR, and KOR, buprenorphine is a DOR and KOR antagonist with high affinity [43]. Buprenorphine is potent at the MOR and the DOR, with efficacy at the MOR, DOR, and KOR in order of descending efficacy [44]. More recent studies of receptor affinity and intrinsic activity in cats have shown that buprenorphine is a MOR, KOR, and NOR receptor agonist and a DOR antagonist [45]. The affinity of buprenorphine for NOR (77 nM) was moderate [43]. The MOR is primarily responsible for analgesic effects, as well as euphoria, miosis, constipation, and respiratory depression [16]. It may have a greater impact at spinal MOR relative to the brain receptors, which is part of what makes buprenorphine classically considered a partial MOR agonist [9]. The DOR has minimal antinociceptive effects relative to the MOR but more activity in chronic pain than acute pain. The DOR also participates in analgesic tolerance and physical dependence [16]. The KOR has been seen to have analgesic and proanalgesic effects due to opioids, while also contributing to miosis and sedation [16].

Buprenorphine’s properties, including low molecular weight, high lipophilicity, and high potency, influence its perceived effects. Potency, the measure of the concentration or quantity of the substance necessary to achieve a predetermined outcome [46], differs depending on the formulation [47]. A value of ten-fold greater than morphine is generally accepted for pharmacoepidemiological research [48]. The drug has a wide tissue distribution and a peak plasma concentration at ninety minutes [17]. Buprenorphine is 96% protein bound after absorption [9]. Oral absorption is considered to be poor because of first-pass metabolism [9]. Transdermal absorption is limited, but there are formulations designed to be more effective. Sublingual administration is considered effective as well [9]. Some studies consider buccal formulations to be the most efficient and have the highest non-intravenous bioavailability [9]. The formulations available for the management of pain show the anticipated routes of administration effects, with parental forms producing the most rapid onset and transdermal forms producing the longest effects [49].

In healthy patients taking buprenorphine/naloxone tablets, they have a peak plasma concentration (T_max_) of 0.75–1.0 h for buprenorphine and 0.5 h for naloxone, demonstrating rapid absorption. Norbuprenorphine plasma concentrations peaked at a T_max_ of 1–1.75 h after the buprenorphine/naloxone tablet administration. The plasma terminal half-life (t_1/2_) was 22–39 h for buprenorphine, 32–44 h for norbuprenorphine, and 1.4–10 for naloxone. Patients who were in withdrawal treatment for opioid dependence had a median T_max_ of 0.75–1 h for buprenorphine, a median T_max_ of 0.75–1 h for norbuprenorphine, and a median T_max_ of 0.5–0.75 [50]. In patients with a history of drug addiction but were drug free at the time of the study, buprenorphine with sublingual and buccal routes had a 51.4% and 27.8% bioavailability, respectively [51].

The half-life is dependent on the method of administration, with 2 h for intravenous, 26 h for the transdermal patch, 28 h for the buccal film, and 37 h for the sublingual tablet [52]. Terminal elimination half-lives were longer for the sublingual and buccal routes of administration than the intravenous route, which may be due to a depot effect from buprenorphine collected in the oral mucosa tissue reservoirs. The time until the maximum concentration occurs was between 0.5 and 3 h sublingually and after 20 min intravenously [18]. Norbuprenorphine had mean peak plasma concentrations that vary by individual and route of administration in healthy patients [18,51]. Intravenous administration of buprenorphine has a 100% bioavailability, buccal has 46–65%, sublingual has 28–51%, and transdermal has 15% [9]. Buprenorphine as a tablet has a bioavailability that is 50–60% that of a buprenorphine solution [53,54]. Intranasal buprenorphine is 50% bioavailable in humans in a polyethylene glycol 300 and 5% dextrose vehicle, with a maximum concentration at 30 min [55]. Buprenorphine’s intranasal bioavailability was 70% with a polyethylene glycol 300 vehicle and 89% with a dextrose vehicle in sheep [55]. The half-life in rats following intravenous administration (2.8 h, [42]) was very similar to humans.

Buprenorphine readily crosses the placenta. However, buprenorphine levels in the third trimester fetal rat brain were only a third of those in the maternal brain [56]. Although there is this notion that norbuprenorphine does not readily cross the blood–brain barrier [9], this may be age or species dependent. Administration of norbuprenorphine (3 mg/kg) to pregnant rats resulted in higher blood and brain levels in the fetus than in the dam [22]. Inhibiting the P-glycoprotein, a drug transporter highly expressed in brain microvessel endothelial cells and placental syncytiotrophoblasts [57], increased rat brain uptake of norbuprenorphine seven-fold [58]. The fetal plasma norbuprenorphine area under the curve was approximately two-thirds that of maternal mice. The fetal AUC of norbuprenorphine glucuronide was three-fold higher than that of the dam. Although interpretation of this study is somewhat limited by analysis of the entire mouse gestational day fifteen fetus (instead of isolating plasma or brain), these findings indicate appreciable fetal exposure to buprenorphine’s biologically active metabolites [59]. Although buprenorphine and norbuprenorphine are transferred into human breast milk, the quantities were low (1%, [60]).

In recent years, contrary to traditional receptor theory, it is clear that different ligands for the same receptor can cause different responses [61]. For receptor theory models to be useful, they must aid in determining the extent to which drug effects can be interpreted and applied to predict future effects [62]. The term “ligand bias” has been used to describe opioid analgesic drugs that elicit a different intracellular response; therefore, their effects are not only the result of receptor binding affinity [44]. Buprenorphine differentiates itself from other opioids in mu-receptor activity, with its slow dissociation from the receptor [63]. Buprenorphine alone is not responsible for its antagonistic effects, but its varying metabolite concentrations through different forms of drug administration may alter the efficacy of the drug.

Traditionally, buprenorphine is described as a partial MOR agonist that is known for limited analgesic effects and developed with the intent for a limited potential for respiratory depression and addiction [15]. However, since buprenorphine’s classification in the 1980s and 1990s, what is known about receptor interaction and activation has changed the meaning of the terms “agonist” and “antagonist” [16,62,64]. Importantly, categories such as full agonist, partial agonist, and antagonist may be unsatisfactory, as a drug’s response may land on a continuum [14]. Reservations regarding buprenorphine’s clinical use were due to misconceptions about an analgesic “ceiling effect” [9]. Until recently, agonists such as buprenorphine have been known for limited intrinsic activity and an inability to produce as large a response at a receptor [15]. Initially, it was concluded that all agonists for a receptor will result in different degrees of the same intracellular response [62,64]. The transduction pathways of a drug activated by an agonist do not act identically for each receptor [4]. Partial agonists are known for their lack of intrinsic efficacy [61]. The antinociceptive effect ascribed to buprenorphine is considered mainly mediated by the MOR [65]. Bell-shaped dose–response curves for buprenorphine in the 1980s and 1990s showed that there is an optimal range of concentrations for a maximum analgesic effect, with a decrease in activity below or above this range [63]. The perception of buprenorphine’s clinical usage may depend on the correct application or interpretation of terms from concepts in receptor theory, such as efficacy and agonist [66].

Studies have suggested that different opioid agonists have different downstream effects in the cell when binding and activating the same receptor. Therefore, different opioids cannot be considered equivalent by changing the dose [16]. It can no longer be assumed that any ligand activating a receptor will produce a response that is relatively the same, with differences attributed to the agonists’ efficacies [4]. Ligands for a receptor can alter the downstream activity in a pathway, known as biased agonism, ligand-directed signaling, and functional selectivity [67]. Opioids that are pure agonists such as morphine or fentanyl produce stronger analgesic effects than drugs such as codeine that have decreased receptor binding [68]. However, factors such as affinity and efficacy, as well as variables such as metabolite binding and concurrent receptor binding may alter the perceived effects and receptor activity of buprenorphine [68].

The binding affinity of buprenorphine and its metabolites to opioid receptors provides the varied effects seen. Binding affinity is the ability of a drug to bind to a receptor and is measured by the equilibrium inhibitory constant (Ki) [9]. Buprenorphine has a high binding affinity at the MOR and KOR, with debated effects [9]. Buprenorphine-3-glucuronide has high affinity for the MOR (Ki = 4.9 ± 2.7 μM) and NOR (Ki = 36 ± 0.3 μM). Norbuprenorphine-3-glucoronide had appreciable affinity for the NOR (Ki = 18 ± 0.2 μM) but not the MOR [69]. Although norbuprenorphine has a greater efficacy, it is considered a less potent partial agonist than buprenorphine at the MOR [70]. A 2002 review described how norbuprenorphine was much less studied than the parent compound but that there was some evidence to suggest that it functioned as a MOR and KOR partial agonist and a DOR and NOR full agonist [11]. Competition assays revealed approximately twenty-five-fold lower norbuprenorpine binding to the NOR than was found with buprenorphine [70]. All metabolites except nubuprenophine-3-glucuronide have analgesic properties [69,71].

Buprenorphine alone is not responsible for its analgesic effects, but its varying metabolite concentration through different forms of drug administration may alter the efficacy of the drug. Norbuprenorphine is one of buprenorphine’s better-studied active metabolites and further research must be performed to understand the other metabolites’ pharmacodynamics [11]. Norbuprenorphine and buprenorphine have substantially different pharmacological profiles. Norbuprenorphine arises as a result of N-dealkylation catalyzed by cytochrome P450 (CYP3A4) in the liver [19,20]. The mechanisms and metabolites of this process are illustrated in Figure 1. At the MOR, both norbuprenorphine and buprenorphine are potent partial agonists, with norbuprenorphine having moderate efficacy and buprenorphine having low efficacy. At the NOR, norbuprenorphine has moderate efficacy and buprenorphine has low efficacy, with both substances having low affinity for the receptor. This information was determined using ligand binding experiments and cAMP assays [70]. Respiratory depression is induced by norbuprenorphine and mediated by the MOR [72]. There is a low risk of respiratory depression with buprenorphine as a monotherapy and this potential effect is rarely considered clinically relevant [73,74]. Buprenorphine’s active metabolite, norbuprenorphine, was ten-times more potent for causing respiratory depression [72]. Buprenorphine was found to be protective against norbuprenorphine’s effect of respiratory depression, both preventing and reversing these effects. An active metabolite of buprenorphine, norbuprenorphine, was alone seen to be responsible for the effects of respiratory depression. Binding experiments show the DOR and, primarily, the MOR as responsible for buprenorphine protecting against norbuprenorphine-induced respiratory depression [17]. The intraventricular administration of buprenorphine and norbuprenorphine showed norbuprenorphine’s analgesic activity was 25% that of buprenorphine [75]. Norbuprenorphine was 50-fold less potent than buprenorphine through intravenous administration and 4-fold less potent after intraventricular administration in in vivo animal studies. This decrease in potency may be due to poor penetration across the blood–brain barrier compared with buprenorphine ([76], although see [21]).

When evaluating results of animal and biochemical studies, norbuprenorphine and buprenorphine are considered by some to be partial agonists at the MOR [76]. The co-activation of the NOR by buprenorphine modulates the antinociceptive effect of buprenorphine at opioid receptors [78]. Additionally, the MOR may be responsible for counteracting the hyperalgesic effect from NOR. If mu receptors are blocked, NORs produce hyperalgesia [79]. Norbuprenorphine, an active metabolite in buprenorphine, has a high binding affinity for the MOR and a low affinity for the NOR and presents as a potent analgesic with an efficacy equal to buprenorphine in the mouse acetic acid writhing test [70]. Buprenorphine’s agonistic effect at the NOR is hypothesized to counter antinociception by buprenorphine and norbuprenorphine on opioid receptors, producing the bell-shaped curves in nociceptive assays [70]. Preclinical reports show that NOR agonism contributes to decreased analgesia at high concentrations. However, buprenorphine’s affinity for the NOR is approximately 50 times lower than its affinity for the MOR and NOR activation, causing a pronociceptive effect that has not been validated in clinical settings [1,69,80]. The agonistic activity and low binding affinity at the NOP receptor contribute to spinal analgesia and may limit the substance abuse potential and tolerance commonly observed with full MOR agonists [81]. NOR antagonists have limited impact on buprenorphine-induced physiological responses in nonhuman primates [82].

Buprenorphine has a bell-shaped response curve for antinociception and catalepsy [28]. The argument can be made that in the clinical setting, the bell-shaped dose–response curve has not been demonstrated for pain [28]. It may be seen at doses that are much higher than typical clinical doses [18]. Buprenorphine was found to be a potent analgesic with full efficacy in mouse models of acute, somatic, and visceral pain. It appears that the analgesic efficacy of buprenorphine is not limited by its categorization as a partial agonist or previous reports of the bell-shaped dose–response curve, as a maximal efficacy for the compound was maintained at almost 100% of the maximal possible effect [83]. In clinical studies, no ceiling has been found with buprenorphine’s analgesic effect [73,74]. Ascending intravenous doses did not produce any ceiling effect up go 0.6 mg of buprenorphine, roughly equivalent to 10–20 mg of intravenous morphine among healthy humans with acute pain [84]. In earlier papers classifying buprenorphine, the reports of the ceiling effect seen with the MOR involved dose ranges that were relatively equivalent to the potency of other drugs it was tested against, such as morphine. A plateau in the dose–effect curve of buprenorphine was identified. However, this team noted that dose comparisons between partial and full mu agonists should be made cautiously, since extrapolation does not accurately estimate potency [85].

Because of the options for different methods of drug administration [9], buprenorphine’s analgesic ability did not appear to be limited and showed promise for pain treatment that was subsequently realized [1,71,86]. Preclinical studies have shown the effectiveness of buprenorphine in various pain conditions [87]. In conscious rats, buprenorphine was even considered 100 times more potent than morphine (equipotent 0.03–3.0 mg/kg s.c) in paw pressure tests; however, buprenorphine produced a bell-shaped dose response curve on the hotplate test. The antinociceptive effects of buprenorphine and morphine were equipotent in both paw pressure and hotplate tests when administered intrathecally at 10 micrograms [88]. The paw pressure test with subcutaneous administration showed buprenorphine was more potent than morphine [89]. The analgesic potency of buprenorphine [89,90] and its lipophilicity and low molecular weight make buprenorphine ideal for transdermal delivery [91]. For this reason, transdermal administration can result in buprenorphine having an increased analgesic efficacy that is 25–50 times more potent than morphine [90]. Lower doses of transdermal buprenorphine were required to produce the same equipotency as transdermal fentanyl [89]. In two case studies, buprenorphine gave a positive response when transdermal fentanyl had failed [89]. Transdermal administration of buprenorphine in chronic non-cancer, neuropathic, and cancer-related pain did not antagonize analgesia and showed beneficial efficacy, safety, or cause withdrawal. Transdermal buprenorphine has been shown to be advantageous for chronic pain treatment [92,93]. Transdermal administration of buprenorphine was efficacious and well tolerated in moderate to severe chronic low back pain [94] and long-term control of chronic pain in cancer patients [87,95]. Transdermal buprenorphine was effective for longer term chronic cancer and noncancer pain, with at least satisfactory analgesic effects reported in 90% of patients [92]. Patients with moderate to very severe chronic pain, both cancer and noncancer related, slept longer uninterrupted by pain; of the 239 patients participating, 90% found satisfactory pain relief and 95% tolerated the patch well [96].

Buprenorphine’s method of administration has implications for the efficacy and clinical benefits or detriments associated with it [71]. Buprenorphine is considered a potent analgesic when administered intravenously, intramuscularly, buccally, and sublingually for moderate to severe pain [86]. Buprenorphine’s slow onset time decreases its effectiveness for acute pain [86]. However, based on the formulation and method of application, buprenorphine can be approximately 25–100 times more potent than morphine [3,89,97]. Intrathecal injections of buprenorphine and morphine showed similar antinociceptive potencies after their peak but with a shallower dose–response curve for buprenorphine. Similar results were shown through subcutaneous administration in the baboon hotplate test [88]. For thermal pain, intrathecal buprenorphine was found to be 17 times more effective than hydromorphone [65]. Buccal administration of buprenorphine was effective and tolerable in opioid naïve patients with moderate to severe lower back pain [98,99] and general “round-the-clock” chronic pain [100]. In a review of thirty-three clinical studies, each trial showed efficacy in buprenorphine for pain relief in the transdermal and buccal forms. Some consider buprenorphine to have the efficacy of a Schedule II drug [101]. Buccal film had and a similar efficacy and tolerance as the transdermal formulation. Buprenorphine buccal film (150–900 μg/12 h) had similar efficacy to hydromorphone hydrochloride (12–64 mg). Sublingual buprenorphine in the tablet form was 15 time more potent than intramuscular morphine. Sublingual buprenorphine is also active longer than morphine [102]. It has been shown to be an effective postoperative analgesic [103,104,105]. The relative potencies of intramuscular to sublingual buprenorphine was 2:1 among postoperative cancer patients [102]. Intramuscular buprenorphine was thirty times more potent than morphine for postoperative pain [106,107,108] and had a longer duration of action than morphine in cancer patients [109]. Oral buprenorphine formulations were twice as likely to be preferred by OUD patients (48.6%) relative to a twice-per-year implant (28.3%) or weekly or monthly injections (23.1%) [110]. Overall, the data from these studies suggest that buprenorphine has equivalent or greater clinical analgesic efficacy than conventional opioids [81].

The classification as a partial antagonist comes in part from the reduced efficacy in morphine and other MOR agonist analgesics when first exposed to buprenorphine. The “antagonist profile” was a conclusion drawn from the reduced efficacy if buprenorphine was injected before morphine. Buprenorphine is still a more potent analgesic than morphine and pentazocine in rat tail pressure tests and is marginally more potent than morphine in mouse and rat tail flick tests [111]. Buprenorphine’s pharmacology allows for it to be combined with other MOR agonists for an additive analgesic effect [87,112]. Administering intrathecal morphine and IV buprenorphine simultaneously alleviates pain with decreased sedation and other side effects than either drug alone [113]. Additionally, switching between buprenorphine and full mu agonists is possible without the loss of analgesic efficacy and without a refractory period when switching from buprenorphine to new mu opioid treatment [86,91]. Overall, practice guidelines state the importance of patients self-reporting effective analgesics, as pain is considered a personal experience that varies based on individual thresholds and tolerances [90].

Opioids rarely bind to a single receptor and will have differences in affinities to others. Buprenorphine co-activates other receptors that may play a role in its efficacy. In a partial agonist, the less than full effect should remain the same, even with full receptor saturation [61]. PET technology shows that buprenorphine can produce analgesia at less than full receptor occupancy, which should make it considered to be a full agonist [66]. Buprenorphine has a high affinity for MOR but occupies fewer receptors for its analgesic effects. Buprenorphine increases MOR expression so that other mu agonists can interact with the receptors [87]. Additionally, buprenorphine’s activation at the MOR occurs at lower levels of receptor phosphorylation [9]. When administering buprenorphine, receptors are available for full agonism at the MOR for the treatment of acute pain [9]. Buprenorphine has antagonistic activity with high binding affinity at the KOR and DOR, which may limit constipation, respiratory depression, dysphoria, and substance abuse [81]. Some of buprenorphine’s negative effects, such as respiratory depression and abuse, can be attributed to peripheral DOR [114]. Although opioid analgesics such buprenorphine often bind to the MOR, there is a variation in their affinity for this receptor as well as their affinity ratio for other receptors, such as the previously mentioned NOR, in addition to the KOR and DOR [16]. Buprenorphine has antagonistic activity at the KOR that causes antihyperalgesic effects to some extent [91]. The antihyperalgesic effects of buprenorphine have successfully treated neuropathic pain [92,97,115], which may show neuropathic pain to be more susceptible to buprenorphine than other opioids [91]. Antagonism from the KOR activation leads to predictions that drugs with lower affinity for the KOR relative to the MOR will be effective in producing MOR-related effects [16]. However, buprenorphine’s KOR activity is debated to be a partial agonist [70,116], antagonist [117], and is even thought to have no activity [90,118]. Some conclude buprenorphine is an KOR antagonist or inverse agonist [9]. Therefore, considering these mixed results [17], the KOR may have a limited contribution to buprenorphine’s activity. Additionally, dimers of the receptors can arise as homo- or hetero-conformations that may have distinct signals [119]. MOR-DOR and DOR-KOR specific agonists have different signals, outcomes, and antinociceptive results [16,112,120].

Despite full MOR occupancy, partial agonism is present in buprenorphine with a partial respiratory effect [73]. Respiratory depression in buprenorphine varies depending upon the method of drug administration. This was significant in animals, even when norbuprenorphine was in greater concentrations than buprenorphine in the plasma [17]. The dose–effect relationship of buprenorphine with respiratory depression suggests limited effects or a plateau of effects over a 0.008–3 mg/kg intravenous dose range [73]. Although the dose–response curve shows a plateau, the idea that the respiratory effects are limited is dangerous, since buprenorphine in combination with drugs such as sedatives can cause fatal respiratory depression [73]. Respiratory effects are rarely reported in maintenance therapy but, in situations of misuse, features of opioid poisoning can be present with buprenorphine [17].

A recent study examined buprenorphine/naloxone as a medication-assisted treatment for kratom use disorder (KUD). Individuals’ dependent on kratom initially sought out the drug due to its opioid-like properties and utilized it as either a therapeutic or for recreational purposes [121]. Mitragynine and 7-hydroxymitragynine make up the composition of kratom. Both compounds, like buprenorphine, have partial MOR agonist and antagonistic effects at the KOR and DOR [122]. The binding affinity of 7-hydroxymitragynine is 13 times more potent than that of morphine [121]. Even with this being a factor, kratom remains unscheduled and can be bought online and in many retail locations in the US. This shows the addictive potential of kratom and the need for treatment. In a case series consisting of 28 patients, all of whom were initially dependent on kratom, 68%, 82%, and 82% showed negative test results for mitragynine at 4, 8, and 12 weeks, respectively, after buprenorphine/naloxone treatment. This report indicates that buprenorphine/naloxone can be an effective treatment option for KUD [122].

## 3. Misuse Potential

A growing number of patients are being treated for OUD in the United States using methadone, injectable naltrexone, and buprenorphine. Buprenorphine is the most widely prescribed for OUD in substance use treatment facilities [123]. Buprenorphine is a Schedule III drug with a unique mechanism of action that has less potential for misuse than Schedule II drugs (e.g., morphine, oxycodone, fentanyl). The lower abuse potential of buprenorphine may mitigate the number of overdose deaths observed with conventional opioids [81]. Buprenorphine is generally perceived to have a low misuse potential alone, particularly when formulated with naloxone [81]. Although human research on buprenorphine misuse is informative, there are interpretive difficulties with self-selected samples that often abuse multiple substances and may differ from the general population on a variety of psychiatric, genetic, and socioeconomic variables. Preclinical research allows causal conclusions.

Two common preclinical methods to assess misuse potential in nonhuman species are self-administration and conditioned place preference (CPP). Buprenorphine could initiate and maintain self-administration in rhesus monkeys with a morphine history and among one who was opioid naïve [124]. Three of four tested baboons intravenously self-administered buprenorphine (1 mg/kg), but at rates half that of codeine [88]. The threshold of brain stimulation reward to the median forebrain bundle was reduced by buprenorphine in rats [125]. Buprenorphine could maintain self-administration in rats but, unlike with other opioids (fentanyl, oxycodone), use did not escalate over time [126].

CPP involves classical conditioning and whether a rodent finds a drug and its associated environment positive, neutral, or negative. Rats formed a CPP to subcutaneous (0.025–0.010 mg/kg) buprenorphine [127]. Wild type mice also showed a CPP to a higher (3 mg/kg) dose but MOR knockouts did not [128]. The combination of diazepam with a buprenorphine dose (1 mg/kg) that was ineffective by itself produced a CPP [129]. There was also a synergistic CPP effect between buprenorphine and cocaine [69].

Two other procedures that provide mechanistic insights into the misuse potential for a drug by targeting the nucleus accumbens, a brain structure important for reward, are microdialysis and fast-scan voltammetry. Microdialysis revealed a doubling, albeit over five-hours, in dopamine from the nucleus accumbens following buprenorphine. The combination of buprenorphine and cocaine produced a larger increase in dopamine than only the cocaine [69]. Voltammetry showed that buprenorphine could produce an intermediate (25%) nucleus accumbens shell response to buprenorphine that was less than that observed with heroin (60%) [130].

There are many studies noting an increasing trend in the misuse, diversion, and self-medication of buprenorphine for withdrawal symptoms [123]. When France introduced buprenorphine in the 1990s (i.e., before the US), they quickly had cases of asphyxic deaths from misuse or concomitant drug ingestion of psychotropic drugs such as benzodiazepines and neuroleptics [17,131]. The Drug Abuse Warning Network has stated that emergency department visits associated with buprenorphine have grown, with a significant amount resulting from nonmedical use [132]. Benzodiazepine prescription was associated with increased risk of opioid overdose, mortality, and, in those using buprenorphine, decreased discontinuation of buprenorphine [123]. Although labelled for sublingual use, buprenorphine was injected by 33% of users [17]. Fatality is typically associated with intravenous misuse through injection of crushed tablets [17]. Among Massachusetts residents, thirty-one percent of the 183 overdoses occurred when individuals used benzodiazepines and buprenorphine [123]. Driving under the influence cases where buprenorphine is implicated almost always also involved other drugs [133].

The general consensus of buprenorphine having low, relative to that of full agonists, misuse potential, should not be confused with an absence of misuse potential [12,13]. The crowd-sourced harm-reduction site Erowid contains reports of buprenorphine and buprenorphine–naloxone recreational experiences [134,135]. Sites such as these provide instructions to dissolve formulations of buprenorphine/naloxone for intravenous use [136]. According to the 2481 submissions to the crowdsourcing site streetrx.com, the street value of buprenorphine/naloxone was only 20% less than that of buprenorphine mono-products. The black-market value of USD 3.95/mg indicates that there is some non-medical use [137]. A prior report using this database found that buprenorphine’s street value was lower (USD 2.13 mg) but still over twice that of another high-potency opioid, methadone (USD 0.96, [138]). Similarly, Maine’s Diversion Alert Program reported on arrests involving illicit or prescription drugs. Arrests for buprenorphine (*N* = 147) exceeded those for oxycodone, hydrocodone, methadone, tramadol, and morphine combined [139].

It is crucial to appreciate that opioids that act on the MOR, whether as full or partial agonists, also increase dopamine. Chronic administration of 3 mg/kg of buprenorphine to rats using an osmotic minipump greatly potentiated the dopaminergic release in the nucleus accumbens of cocaine [140]. Further, escalating doses of buprenorphine to mice decreased striatal D_1_ and D_2_ receptors [141]. Although there would be value in a direct comparison of the potency of buprenorphine and NBUP for self-administration and CPP in rodents, any effort to develop an “abuse-proof” formulation of BUP will be unsuccessful. With the development of new buprenorphine analogues, the goals will continue to be decreasing the harms relative to those produced by the misuse of illicit or licitly produced heroin, fentanyl, or other opioids.

## 4. Toxicology

Buprenorphine has adverse effects that are similar to other opioids and produces dizziness, nausea, vomiting, sedation, respiratory depression, and constipation. It produces more sweating than codeine, dextropropoxyphene, oxycodone, and pentazocine [18]. For obvious ethical reasons, controlled studies of lethality are only completed with experimental animals. The acute toxicity (LD_50_) of buprenorphine varies based on the method of drug administration (see Table 1, from [32,33,34,35]). When comparing norbuprenorphine and buprenorphine through intravenous administration, the LD_50_ values are 146.5 and 234.6 mg/kg, respectively, and the norbuprenorphine-to-buprenorphine LD_50_ ratio was found to be 1/16–1/23 [17].

The dose–effect relationship of buprenorphine with respiratory depression suggests limited effects or a plateau of effects over a 0.008–3 mg/kg intravenous dose range [73]. In a study with healthy volunteers, intramuscular buprenorphine (0.15–1.2 mg) increased the risk of respiratory depression linearly; however, the effect was not clinically significant [142]. With sublingual buprenorphine (1–31 mg), patients reached respiratory depression at doses of 8 mg or more [85]. A study on 50 postoperative patients with intravenous buprenorphine (0.4–7.0 mg) showed no signs of respiratory depression for a 24 h period [143]. Healthy volunteers with intravenous buprenorphine (0.1 mg/70 kg body weight) demonstrated a ceiling in respiratory depression but not in analgesic efficacy [52]. Animal experiments show that the respiratory ceiling occurs at a lower dose (>0.2 mg/ kg) than the analgesic effect ceiling, which will only occur in doses beyond the therapeutic dose range [74,84]. The respiratory effects are rarely reported in maintenance therapy; however, in situations of abuse, features of opioid poisoning can be present with buprenorphine [17]. Experimental and clinical data show that there is a limit of buprenorphine’s maximum depressant effect [91]. Although the dose–response curve shows a plateau, the idea that respiratory effects are limited is dangerous, since buprenorphine in combination with drugs such as sedatives can cause fatal respiratory depression [73].

Importantly, studies have shown that buprenorphine treatment can be effective in reducing mortality for individuals with OUD and investigations support that ceasing these opioid agonist treatments can lead to higher rates of all-cause mortalities [144]. Evidence suggests a protective reaction to fentanyl-induced respiratory depression at 2 ng/mL concentrations and higher. Furthermore, when the MOR occupancy by buprenorphine is sufficiently high, fentanyl is not able to bind and activate the MOR, resulting in a decrease in respiratory depression in those overdosing on fentanyl [145]. Buprenorphine, in turn, can be the cause and treatment for respiratory depression. Buprenorphine brings about mild respiratory depression, whereas at high doses fentanyl causes significant respiratory depression and apnea [145]. Although buprenorphine has been observed to cause partial respiratory depression, the results indicate that administration of buprenorphine buccal film may have a decreased risk of abuse and respiratory depression compared with the full MOR agonist oxycodone [146].

Buprenorphine has an increased potential for misuse when central nervous system depressants such as benzodiazepines are used simultaneously [123]. Benzodiazepines are not CYP3A4 inhibitors; however, some, such as diazepam and flunitrazepam, are metabolized through this enzyme. This drug interaction is likely additive or synergistic [21]. Interactions between benzodiazepines and opioids, as well as buprenorphine and methadone, have resulted in respiratory depression in animal models and humans [17,147]. Opioids and benzodiazepines act in combination with different classes of opioid and GABA receptors, but only limited interactions have been reported [17]. Benzodiazepine and buprenorphine’s concurrent use causes a decreased reaction time and is associated with an increased risk for emergency room visits for accidental injury [123]. It should be noted that many patients with substance use disorders use benzodiazepines during treatment [17]. However, whereas one-third of patients are prescribed both buprenorphine and benzodiazepines, approximately another third regularly use illegally obtained benzodiazepines, making it difficult to decrease the risk of substance use relapse. Because of the concern that benzodiazepines might impede opioid maintenance therapy, the US Food and Drug Administration urged withholding opioid agonist treatment if the patient was taking benzodiazepines [123]. Pharmacodynamic interaction is the expected cause of buprenorphine–benzodiazepine drug interaction found in humans and animals. However, flunitrazepam–buprenorphine drug interaction is thought to have a pharmacokinetic interaction. Flunitrazepam alters buprenorphine lethality in rats, with a six-fold decrease of its LD_50_, which appears to be opioid-specific as there was only a two-fold decrease in methadone and no significant effect on morphine [17].

Some studies show here is a significant amount of norbuprenorphine remaining in plasma following buprenorphine’s administration [148], which is contrary to other reports [75]. Reported buprenorphine overdoses in the mid 2000s can be related to varied norbuprenorphine plasma concentrations [149], which could be related to method of administration [17,51]. Buprenorphine’s clearance in anesthetized patients was seen to be lower than in individuals not under anesthesia, as well as in patients with reduced hepatic blood flow as a result of another administered anesthetic [105]. Fatal cases related to buprenorphine have had high plasma or tissue concentrations of norbuprenorphine, suggesting its role as a respiratory depressor may be a significant future consideration in buprenorphine’s toxicity. Significant respiratory depression has been found in rats with a single intravenous administration of 3 mg/kg norbuprenorphine. The mechanism for the respiratory effects from buprenorphine is unknown and is not responsive to naloxone [17,21]. However, contradictory reports have been noted, where patients show improvement using 0.4–0.8 mg of naloxone [17]. It is still unclear if norbuprenorphine alone is significant enough to be the cause of buprenorphine-related death [17]. Respiratory depression in buprenorphine varies depending upon method of drug administration. Despite full MOR occupancy, partial agonism is present in buprenorphine with a partial respiratory effect [73]. This was significant in animals, even when norbuprenorphine was in greater concentrations than buprenorphine in the plasma [17]. The dose–effect relationship of buprenorphine with respiratory depression suggests limited effects or a plateau of effects over a 0.008–3 mg/kg intravenous dose range [73]. Although the dose–response curve shows a plateau, the idea that respiratory effects are limited is dangerous, since buprenorphine in combination with drugs such as sedatives can cause fatal respiratory depression [73]. Respiratory effects are rarely reported in maintenance therapy; however, in situations of abuse, features of opioid poisoning can be present with buprenorphine [17]. It would be difficult to understate the importance of polysubstance use among decedents where buprenorphine was identified [150]. An investigation of 117 fatalities that tested positive for buprenorphine found that benzodiazepines and neuroleptics were also present [131]. A report from Rhode Island on opioid overdose cases that were positive for buprenorphine discovered that the average number of drugs and metabolites on toxicology testing was nine [151].

Poison control reports involving buprenorphine increased by 67% from 2011 to 2016 [132]. A younger age may increase the sensitivity to buprenorphine-induced respiratory depression, which can be severe enough to require naloxone treatment. Some overdoses involved an orange, hexagon-shaped, lemon–lime flavored sublingual tablet [152], whereas others reported an intravenous formulation for veterinary use [153]. Buprenorphine was placed on the list of drugs that can be fatal with a single dose for a 10 kg toddler [154]. There have been case reports of child fatalities following accidental buprenorphine/naloxone exposure. A ten-month-old infant was found unresponsive eight hours after a family member removed a sublingual buprenorphine/naloxone (8 mg/2 mg) pill from his mouth. Postmortem toxicology revealed a serum concentration of 52 ng/mL buprenorphine and 39 ng/mL norbuprenorphine but he tested negative for other illicit, prescription, or over-the-counter drugs [155]. Among the over eleven-thousand reports from 2007 to 2016 involving buprenorphine to US poison control centers involving children and adolescents (age ≤ 19), the vast majority involved children younger than six (86.1%) and were unintentional (89.2%) [156]. A controlled study in infants and toddlers found that buprenorphine (1.5 or 3 μg/kg) produced a greater respiratory depressant effect than morphine (50 or 100 μg/kg, [157]). Findings such as this have prompted calls for improved education on safe buprenorphine storage in a locked medicine cabinet or storage box in its original container and disposal [158] and a greater use of child-resistant packaging [159].

## 5. Buprenorphine: Human/Clinical Use

Buprenorphine is traditionally used for OUD and is expanding into use in acute and chronic pain disorders [160]. Buprenorphine’s adult dosing is typically 8 mg as a sublingual tablet [160]. When combining buprenorphine and naloxone, this dose can be 4.2 mg/0.7 mg as a buccal film. For OUD, there are sublingual daily formulations (1–4 mg), weekly or monthly extended-release injections (ranging from 100–300 mg monthly), and subdermal implants (2 mg/day). For the purpose of acute pain, 0.3 mg can be administered intramuscularly or through slow IV every 6–8 h as needed [160]. Chronic pain typically uses the buccal film or transdermal patch formulation [160]. The suggested dose varies according to the patient’s previous experience with opiates [160]. Opioid-naïve patients and opioid non-tolerant patients can have 75 mcg daily or every 12 h for four or more days, which can then be increased to 150 mcg every 12 h [160]. Patients converting from other opioids need to be removed from all other opioid when buprenorphine is started; then, the initial dose is based on the patient’s prior daily opioid dose (less than 30 mg of oral morphine equivalents would receive 75 mcg buccal buprenorphine daily or every 12 h, 30–89 mg of oral morphine equivalents receive 150 mcg every 12 h, 90–160 mg oral morphine equivalents receive 300 mcg every 12 h, and those receiving over 160 mg of oral morphine equivalents should consider an alternate analgesic use) [160]. For the transdermal patch, in opioid-naïve patients, 5 mcg/h must be applied once every 7 days. Patients converting from other opioids need an initial dose based on the patient’s prior daily opioid dose [160]. Less than 30 mg of oral morphine equivalents would receive a 5 mcg/h buprenorphine patch every seven days, 30–80 mg of oral morphine equivalents receive 10 mcg/h patch every seven days, and over 80 mg oral morphine equivalents receive 20 mcg/h every seven days, but this may not be adequate and clinicians should consider an alternate analgesic use [160].

Buprenorphine has benefits in special populations including the elderly and those with renal or liver impairment [1]. Buprenorphine’s pharmacokinetics does not change with old age when comparing individuals over 70 years with younger individuals (average 32 years old) [161]. In patients with renal failure, norbuprenorphine does not accumulate and therefore buprenorphine can be administered at normal doses [162]. Buprenorphine has less sedation, cognitive impairment, and risk of falls for the elderly population, making it an ideal first-line opioid analgesic in the elderly population [163]. Mild to moderate liver impairment does not require dose adjustments for buprenorphine due to upregulation of UGTs in remaining hepatocytes and extrahepatic glucuronidation [1,164]. It should be noted premature infants and neonates experience delays in clearing buprenorphine due to delays in CYP3A4 expression. Buprenorphine is not associated with post dialysis. Glucorindation is delayed in premature and low-birthweight babies as well [1]. Additionally, buprenorphine has less binding to the gastrointestinal MOR, reducing the constipation risk expected with other opioids.

In 2017, the Canadian Agency for Drugs and Technologies in Health published a summary of all studies to date that involved buprenorphine in chronic pain management [165]. The review found that tramadol, codeine, and buprenorphine produced similar analgesia in osteoarthritis. Fentanyl and buprenorphine produced similar analgesia in neuropathic pain and pain from AIDS. Transdermal buprenorphine at 20 μg/h was equivalent to 40 mg μg/h oxycodone daily when treating lower back pain [165]. However, direct analgesic equivalents may not be generalizable across pain populations (chronic lower back pain, neuropathic pain, postoperative pain, cancer pain) [1]. In 2019, an expert panel provided clinical recommendations for discussing buprenorphine for chronic pain. Buccal film and transdermal formulations of buprenorphine are indicated for pain management for conditions that require daily, continuous opioid treatment [146]. Patients with chronic pain may benefit from buprenorphine if they have an increased risk of adverse events typically relating to opioids, including those suffering from obstructive sleep apnea, a comorbid psychiatric diagnosis, pulmonary disease, or those who have a high body mass index [146].

Buprenorphine can be used in transdermal and parenteral formulations for postoperative pain. Doses of transdermal buprenorphine between 5 and 20 μg/h produced the same analgesic effects as 150–300 mg/day of tramadol in single-level spinal surgery [155]. The time to activation of patient-controlled analgesia was longer for buprenorphine than morphine in one study. In another study, buprenorphine had more frequent demands for dosing in the first 6 h after surgery but the same number of rescue doses as morphine after 6 h [166].

Buprenorphine is an ideal option for those with OUD and expands the treatment options for those with the condition. The low ceiling on agonist effects and slow dissociation from the MOR reduces likelihood of abuse, favoring its use in OUD relative to methadone [167]. Physicians surveyed in 2017 cited a “lack of belief in agonist treatment” as a reason for choosing not to prescribe buprenorphine [167]. There is no observed ceiling on the analgesic effect in clinical studies [168].

## 6. Prenatal Correlates and Consequences

Methadone or buprenorphine, in combination with counseling and behavioral therapy, are currently considered first-line treatments for pregnant women with OUD [169]. The cellular targets of buprenorphine and norbuprenorphine (KOR, MOR, NOR) may serve different functions during development [5,6] than they do in the adult brain. Pregnancy increases the activity of CYP3A4 and the glucuronosyltransferase enzyme superfamily, resulting in a lower AUC of buprenorphine and norbuprenorphine [170,171]. The actions of glucuronosyltransferase are highlighted in Figure 2. Women may need even higher dosages during pregnancy to account for the increased activity of the enzymes responsible for buprenorphine and norbuprenorphine metabolism [166,167]. Controlled behavioral teratology studies in humans are not ethically permissible and even the descriptive reports are often challenging to interpret due to many confounding factors including often under-reported polysubstance misuse and other socioeconomic disparities. There are some preclinical reports, primarily in rats, that have identified abnormalities in a wide variety of structural and functional endpoints following prenatal buprenorphine exposure. Mice have atypical buprenorphine pharmacokinetics including only modest amounts of norbuprenorphine crossing the blood–brain barrier [172,173,174], which is unlike sheep [23]. Another possibility that could account for the differences among studies is that analytical chemistry procedures have been refined. The selection of dosing regimens for use in rats, with their proportionally larger livers, faster metabolisms, and shorter pregnancies (three weeks), that are clinically relevant to humans is not trivial [175]. Maintenance doses of 3-24 mg per day are employed in a 70 kg person, which translates to 0.04 to 0.34 mg per kg body weight, although doses of 0.41 have been reported [60]. The rat doses of 0.3 to 3.0 mg/kg that are often administered subcutaneously or by implantable mini-pumps, may therefore be considered as clinically relevant. A minor caveat is that the mini-pumps release a fixed dose per day but, on a mg/kg basis, this may decrease somewhat as the maternal body weight increases during pregnancy [176]. Oligodendrocytes make myelin and express opioid receptors. The prenatal and early postnatal stages correspond to third-trimester human brain development and buprenorphine (1 mg/kg per day) from day seven of gestation until postnatal day twenty-one decreased myelinated axons at twenty-six days of age in the corpus callosum [177]. A subsequent investigation determined that prenatal and early postnatal treatments with 1 but not 0.3 mg/kg reduced the overall brain mass in males and females at postnatal day 21 [178]. This dosing regimen reduced brain-derived neurotrophic factor and biochemical parameters of neural stem and progenitor cells [179]. A 0.5 mg/kg dose to pregnant rats for one week induced a subtle but statistically significant (18%) down-regulation in MOR binding in the brain of dams but a 64% decline in the offspring, indicating that the developing brain was more sensitive [180].

A meta-analysis of the six rat investigations that studied brain function determined that perinatal buprenorphine impacted a variety of domains including rodent models of emotion, cognition, and responsiveness to addictive drugs [182]. Prenatal exposure to 1 but not 0.3 mg/kg buprenorphine increased immobility, independent of sex, in the forced swimming and tail suspension tests [178]. This profile of behavior was replicated, albeit by the same laboratory, and is interpreted as an increase in depression-like behavior [179,183]. Young adult rats that had received buprenorphine perinatally (1 mg/kg) showed a pronounced deficit in recognition memory [176]. The view that 1 mg/kg was the threshold to produce neurobehavioral effects was challenged by a recent investigation that identified decreased maternal care, delays in offspring maturation, and abnormalities in response to a painful stimuli at the 0.3 mg/kg dose [184].

Given that the neurostructural (e.g., decreased myelination [6] and nerve growth factor [185]) and behavioral effects were causally linked to perinatal buprenorphine in rats and that this is a rapidly expanding research area, it is important to verify these results in humans. Unfortunately, this is an extremely difficult area to study. The principle challenge is that although buprenorphine is an evidence-based intervention to reduce other opioid use [186], it does not eliminate it. For example, of nine women receiving buprenorphine who completed thrice weekly urine testing during pregnancy, all tested positive at least once for other opioids. An average of one-quarter of samples were positive, but this ranged from 4 to 75% [26]. Untangling the contribution of illicit and prescription opioids to maternal and fetal outcomes is not trivial [175]. Other challenges include determining the involvement of prenatal alcohol, nicotine, maternal stress and diet, prenatal care, or the postnatal environment. If that were not enough, this can be an inherently challenging population and treatment retention is lower with buprenorphine than with methadone [186]. Despite these many caveats, a meta-analysis of the offspring of mothers receiving opioid maintenance during pregnancy showed lower scores relative to an unexposed comparison group for vision (effect size = 0.25), motor activity (0.37), attention and executive functioning (0.40), and psychomotor function (0.56). The magnitude of the overall effect size (0.49) was equivalent to a seven-point IQ deficit [182]. The literature in this area is not sufficiently advanced to differentiate whether methadone or buprenorphine is correlated with more adverse outcomes [187]. Given the increasing availability of buprenorphine [188], this should be a high priority area for further study using clinically relevant dosing regimens and endpoints (e.g., structural neuroimaging) in rodents. There is currently only limited research that has evaluated the unique contribution of direct administration of norbuprenorphine in experimental animals to adverse effects (e.g., neonatal opioid withdrawal syndrome) [22,189].

## 7. Conclusions

Buprenorphine is widely used for OUD, including during pregnancy, and pain. Distinctions between “weak” and “strong” opioids or “full” and “partial” agonists may be needed to account for “weak” opioids such as buprenorphine having characteristics considered “strong” [16]. This is ineffective if used clinically, as “weak” opioids are considered less likely to lead to addiction and adverse side effects, whereas this can be seen as untrue in buprenorphine. Reducing diversion may require developing new misuse-deterrent formulations of buprenorphine [132]. More work needs to be conducted in determining drug interactions with buprenorphine that are the result of some interaction or inhibition with metabolizing enzymes such as CYP3A4 and CYP2C8. Synergistic or additive effects due to other opioids, alcohol, and neuroleptics should be considered [17]. Preclinical investigations are necessary to examine age differences in sensitivity to adverse effects such as respiratory depression. The brain structure [177] and functional abnormalities in rats following relatively low perinatal doses [178,183,184] are concerning and may warrant follow-up (e.g., diffusion tensor imaging to evaluate white matter integrity in buprenorphine-exposed nonhuman primates or children). Clinical studies are needed to determine the ceiling for analgesia in humans and the dose it occurs at [73]. Additionally, without taking into account the full effects of the metabolites’ transduction based on method of administration, there can be adverse effects as a result of the residual effects of buprenorphine’s metabolism. The bioavailability of certain metabolites in plasma, such as norbuprenorphine, requires more research as this has implications for medications that can be co-administered with buprenorphine. Without taking into consideration factors such as the method of administration, this can lead to incorrect assumptions in the efficacy of the opioid prescribed. Additionally, without taking into account the full effects of the metabolites’ transduction based on method of administration, there can be adverse effects as a result of the residual effects of buprenorphine’s metabolism. Some information sources [13] describe the mechanism of action of buprenorphine as simply a mu partial agonist and provide limited information about the major, and biologically active, metabolite norbuprenorphine, which may be a considerable oversimplification. Continued efforts to better understand the complex pharmacodynamics and pharmacokinetics of buprenorphine and its metabolites will result in a better appreciation of the benefits and risks of this ubiquitous opioid.

## Figures and Tables

**Figure 1 pharmaceuticals-16-01397-f001:**
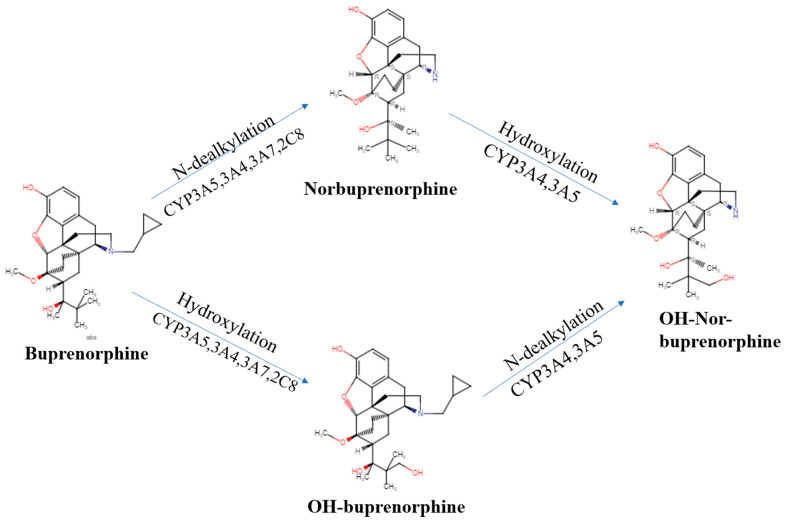
Buprenorphine, norbuprenorphine, and its metabolites utilizing the Reaxys database [77].

**Figure 2 pharmaceuticals-16-01397-f002:**
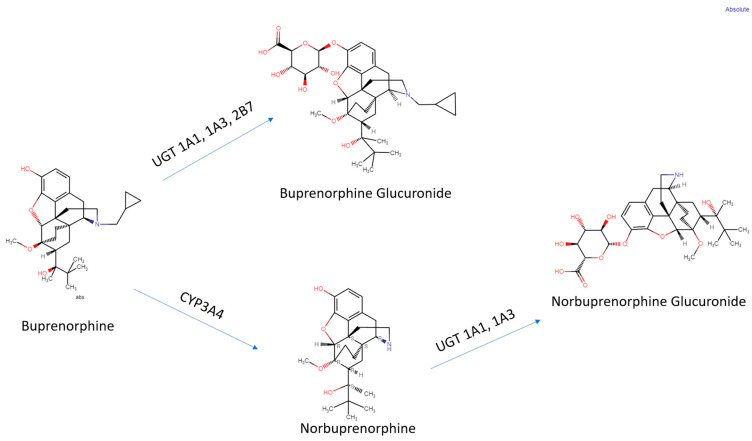
Buprenorphine and norbuprenorphine glucuronidation. UGT is the glucuronosyltransferase enzyme [181].

**Table 1 pharmaceuticals-16-01397-t001:** Buprenorphine pharmacokinetic parameters including half-life, time to maximum concentration and area under the curve by route of administration [32,33,34,35].

Route of Administration	Brand Name	T_1/2_	T_max_ (h)	C_max_ (ng/mL)	AUC (h × ng/mL)
Sublingual (2 mg)	Subutex	31.7 ^B^ 39.3 ^NB^	1.8 ^B^ 2.4 ^NB^	1.3 ^B^ 0.3 ^NB^	10.9 ^B^ 12.4 ^NB^
Buccal film	Belbuca	27.6 ± 11.2	3.0	0.17 ± 0.30	0.46 ± 0.22
Transdermal (at 10 mcg/h, steady state)	Butrans	26	-	0.224	27.543
Injectable ^ER^ (300 mg)	Sublocade	Days 1032–1440	-	10.12	3006

T_1/2_: half-life; T_max_ (h): time to maximum concentration; C_max_: maximum concentration; AUC: area under the curve by route of administration; ^B^: Buprenorphine; ^ER^: Extended Release; ^NB^: Norbuprenorphine: ^NB^ [32,33,34,35].

## Data Availability

Data sharing is not applicable.

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
