# Peer review of "An Examination of the Complex Pharmacological Properties of the Non-Selective Opioid Modulator Buprenorphine"

_pharmaceuticals, 2023, doi:10.3390/ph16101397_

Round 1

Reviewer 1 Report

This manuscript presents an interesting review on buprenorphine use  in human.

The paper content is I think quite complete. But the authors should better differentiate the results obtained in human  from those obtained in animals. They often jump from human to animal without transition. This brings some confusion.

It is clear that you cannot do certain studies in vivo in human. Thus you must use a model.  But explain!

Thus the manuscript should be modified to make clear huma versus animal.

Sometimes also the mode of administration is mixed so that the reader has difficulties in understanding.  Could you better explain the different administrations, and their use in pain treatment or recreational use? Could you put a diagram or a figure to explain the bell shaped effect.

I think you must add a scheme showing the structure of buprenorphine and its metabolites. This would be clearer.

For a non specialist reading your review should bring informations on how to handle a normal and anormal usage of buprenorphine.

As it is the manuscript cannot reach this objective.

I note her a few remarks I have done reading the text:

Page 2 paragraph 2 : I think a scheme of metabolization would be a nice addition.

Also You speak about human PK then jump directly to sheep. It would be nice to make a transition explaining that jump. In the following paragraph I do not differentiate what is human or sheep?

Page 3 second paragraph : This paragraph is confusing for a non specialist. Could you clearly separate the effect for each metabolite? What is the meaning of potency here: make that clearer.

Rest of the introduction : You have the material for a great review. But you must absolutely make clear what is human and separate from data obtained with sheep or rats. Then you could try to tell the consequences for human.

For instance Use transition words like However, Although ... This would make your point clearer.

Line 236 : You should name that P450  CYP3A4 (a major human  P450)

Since CYP3A4 is often inhibited by a number of drugs, there could be drug interaction (DDI) with some drugs rtaken by the patient. For instance Ritonavir (in HIV treatment) some antibiotics like erythromycin derivatives a..., This point is treated later

Alltogether I think that the material in your review is quite complete and it must be a little rearranged to help the reader understanding what is human, what is animal

How one use this buprenorphine in real life.

Then the review will be great.

Author Response

Hello, 

I want to thank you for your edits. I apologize for the late resubmission, however, due to many factors, we faced many challenges in completing the edits in a timely manner. The last author is a Stage IV kidney cancer patient who has received radiation therapy (twice) and is currently on immunotherapy and some other medications (tyrosine kinase inhibitors) that reduce productivity. The first author had to complete shelf exams only last week and the middle author just started an MD program!  Despite multiple generous extensions by Managing Editor Edith Fang, we ran out of time. 

Despite challenges we have put an immense amount of work into the paper and the edits we received. 

Thank you for your consideration.

Reviewer 2 Report

The authors present a review report as to the pharmacological properties of buprenorphine. As the authors state there is a lot of background literature (refs 2, 8-14) and they elaborate that their work aims to provide some updated details. Although it is not clear what the added value of previous works is given (tables and figures missing) the work is mostly a detailed description about the drug as to its PK/PD, toxicity, prenatal studies as well as its misuse. Overall, the work is interesting to read in most parts with a lot of information in it. However, the lack of information in figures and tables makes it more of a encyclopedic and less a scientific article. There are some comments that should be addressed though. 

Major comments:

1. Figures 1 and 2 are not standing alone and legends are more informative than the figures. The sketches are not informative.

Minor comments: 

1. Consider adding a figure showing the metabolic pathways of buprenorphine and its metabolites

2. Consider creating a table with the PK parameters of buprenorphine. 

3. Exclude phrases "some believe" (line 68) or later "unusual (line 100) since they are not scientifically sound.

4. Line 100-106 and line 118-122. Same phrases these two sections need rephrase. 

5. Line 134 needs references. 

6. The statement in lines 213-215 needs references.

7. Line 467 Results?

Thank you for offering me to review this work. I am not convinced about the novelty or the added value to the field but it could be of interest for the readers. Moreover, its structure is more of an encyclopedic and less of a scientific article. It has a lot of similairities with ref 9 especially as to the title. There are also some comments that need to be addressed but the work generally can be considered for publication as a review article. 

Author Response

(The authors gave the same response as above.)

Reviewer 3 Report

In this review, Pande and colleagues extensively describe the interaction of buprenorphine and norburenorphine with various mu opioid receptors, the variations in pharmacokinetics and metabolism of buprenorphine due to route of administration, and  the potential for side effects and toxicology due to both parent drug and metabolite.

While this is a very detailed review, the lack of organization of topics and failure to group the information into sections with defined objectives and conclusions decreases readability.  Some suggestions include

1) Include information about current use, routes and effects in the introduction.  Do not assume the reader understands how the drug is typically used.  Many who read the report may be basic scientists who use buprenorphine in animals for pain management and may not understand its use in humans.  

2) Subdivide the pharmacokinetic and pharmacodynamic section further into their own sections.  

3) For the PK section, a figure outlining metabolic pathways would be helpful as would a table of PK parameters and how they differ by route of administration.

4) For the PD section, please discuss the activity of buprenorphine and norbuprenorphine separately and on each receptor separately.  Then once that information is detailed, discuss potential implications and effects of the simultaneous presence of both compounds, including a discussion of partial agonism and antagonism.  A table describing potency for each receptor, hypothesized effects due to signaling through that receptor, classification as agonist or partial agonist or antagonist, etc. would be helpful.

5) At end of each of the PK and PD sections, a subsection could be included to describe interaction of buprenorphine and norbuprenorphine with other opioids.  

6) Subsequent sections on misuse potential, toxicology, and prenatal correlates and consequences should be shortened and focused more directly on each topic.

Minor points

1) There is a section entitled "Results" which just includes the two figures.  It is assumed this is a typo to have this subheading?

2) There is significant repetition of facts and ideas.  The manuscript should be edited to remove this repetition.  For example lines 118-126 are very repetitious of lines 100-116.

Author Response

(The authors gave the same response as above.)

Round 2

Reviewer 3 Report

In this resubmission, Pande and colleagues have addressed many of my previous suggestions, including 

1) addition of information on current use, including routes, of buprenophine in humans.

2) inclusion of a figure showing limited metabolic pathways

3) addition of more details on interaction of buprenorphine and norbuprenophine with other opioids and drugs and more details of activity of each compound on the various receptors.

However, given the complexity of the subject mattered handled, the article still suffers from a lack of clarity and readability.  

The article would still benefit from some additional tables and figures to better organize the information, as I and the other reviewers have suggested. 

Some specific thoughts include

1) Add additional details to the metabolic pathway figure including glucuronidated metabolites.

2) Include details of norbuprenorphine PK in the table showing buprenorphine PK.  Discuss how route of administration may impact generation of norbuprenorphine and subsequent effects.

3) Include a table with the most widely accepted views of agonist and antagonist activity for buprenophine and norbuprenophine on each of the opioid receptors.

Minor:

lines 220/221 repetitive

Lines 222-228 - Provide information about why buprenophine is contraindicated with these other other drugs,

Table 1 - keep units consistent please
